# Solution-Processed CsPbBr_3_ Perovskite Photodetectors for Cost-Efficient Underwater Wireless Optical Communication System

**DOI:** 10.3390/mi15101185

**Published:** 2024-09-25

**Authors:** Jiakang Wei, Yutong Deng, Jianjian Fei, Tian Yang, Pinhao Chen, Lu Zhu, Zhanfeng Huang

**Affiliations:** 1Guangdong Provincial Key Laboratory of Optoelectronic Information Processing Chips and Systems, School of Microelectronics Science and Technology, Sun Yat-sen University, Zhuhai 519082, China; weijk5@mail.sysu.edu.cn (J.W.); dengyt35@mail2.sysu.edu.cn (Y.D.); feijj@mail2.sysu.edu.cn (J.F.); yangt253@mail2.sysu.edu.cn (T.Y.); chenph8@mail2.sysu.edu.cn (P.C.); zhulu5@mail.sysu.edu.cn (L.Z.); 2School of Electronics and Information Technology (School of Microelectronics), Sun Yat-sen University, Guangzhou 510006, China

**Keywords:** perovskite, photodetector, underwater wireless optical communication, visible light communication, CsPbBr_3_

## Abstract

Underwater wireless optical communication (UWOC) has attracted increasing attention due to its advantages in bandwidth, latency, interference resistance, and security. Photodetectors, as a crucial part of receivers, have been continuously developed with the great progress that has been made in advanced materials. Metal halide perovskites emerging as promising optoelectronic materials in the past decade have been used to fabricate various high-performance photodetectors. In this work, high-performance CsPbBr_3_ perovskite PDs were realized via solution process, with low noise, a high responsivity, and a fast response. Based on these perovskite PDs, a cost-efficient UWOC system was successfully demonstrated on an FPGA platform, achieving a data rate of 6.25 Mbps with a low bit error rate of 0.36%. Due to lower background noise under environment illumination, perovskite PDs exhibit better communication stability before reaching a data rate threshold; however, the BER increases rapidly due to the long fall time, resulting in difficulty in distinguishing switching signals. Reducing the fall time of perovskite PDs and using advanced coding techniques can help to further improve the performance of the UWOC system based on perovskite PDs. This work not only demonstrates the potential of perovskite PDs in the application of UWOC, but also improves the development of a cost-effective UWOC system based on FPGAs.

## 1. Introduction

The scope of human exploration continues to expand, and the oceans covering about 70% of the Earth’s surface are becoming a key focus for future exploration. Underwater communication is generally divided into wired and wireless types, and plays an important role in ocean exploration. Wired communication uses cables or optical fibers to provide stable communication media, which face issues such as difficult maintenance and low flexibility [1]. In contrast, underwater wireless communication avoids these issues and has a minimal impact on the marine ecosystem. Among various underwater wireless communications, acoustic communication is a well-developed and widely adopted technique. The data rate of underwater acoustic communication ranges from tens of bits per second (bps) to tens of kilobits per second [2,3]. However, acoustic communication using sound as a medium has the disadvantage of having a small bandwidth, high latency, and poor security [4,5]. In addition, acoustic communication systems have large and high-power transceivers [6]. Underwater wireless optical communication (UWOC) has attracted much attention due to the advantages of real-time transmission, a large bandwidth, good interference resistance, good security, a small transceiver size, and low energy consumption [7]. Recently, much effort has been made to improve the performance of UWOC regarding light sources and modulation in transmitting end [8,9], the influence of water [10,11], photodetectorsin the receiver [12,13], multiple-input multiple-output configurations [14], and signal processing [15,16]. 

Field-programmable gate arrays (FPGAs) have become crucial hardware platforms in modern communication systems, due to their flexibility and efficiency. The programmability of FPGAs allows designers to dynamically configure hardware functions for quickly adapting to different communication protocols and standards. In visible light communication systems, the parallel processing capabilities of FPGA favor high-precision ADC data transmission, providing effective data processing without custom integrated circuits [17,18]. Several groups have demonstrated UWOC systems using FPGAs [18,19,20]. Marek Doniec et al. from the Computer Science and Artificial Intelligence Laboratory have demonstrated a UWOC system based on FPGAs, and achieved a data rate of 1.2 Mbps at distances of up to 30 m in clear water [21]. Later, they achieved a real-time video delivery with 15 frames per second of 1288 × 964-pixel images based on the updated UWOC system with a 4 Mbps bandwidth [22]. Xiao Li et al. achieved a data rate of at least 50 Mbps with a very low bit error rate (BER) over an underwater channel of 1 m [19]. Even though the data rate is far lower than that of offline systems at the gigabit level [23,24], UWOC systems based on FPGA boards, without using expensive and complicated equipment, show huge potential in their light weight and low cost, and this is also a critical step for achieving real-time communication [20].

Sea water as a communication medium has a low-loss window in the blue–green light spectrum (450 nm–550 nm) [25], in which an optical signal has a minimum attenuation and the largest transmission distance. All inorganic CsPbBr_3_ perovskite possessing outstanding optoelectronic properties (high absorption coefficient, high carrier mobility, etc.) [26] are promising candidates for optical receivers in UWOC systems. The response spectrum of CsPbBr_3_ perovskite (bandgap about 2.3 eV) photodetectors (PDs) exactly covers the propagation window of sea water, intrinsically preventing the impact of long-wavelength light. Perovskite PDs have been reported for their application in UWOC previously [27,28,29]. Jiang Wu et al., using a Br-terminated MXene-treated perovskite PD as a receiver, demonstrated UWOC with data rate of 128,000 baud per second [28], and a long response time of 29.9 μs was the limitation of the data rate. Liang Shen et al. employed an ultralow dark current and self-powered perovskite PD as a receiver in an UWOC system and demonstrated a transmission rate of only 300 bps. Improving the response speed and reducing the noise current of perovskite PDs is of great importance to achieve a high speed of data transmission. Recently, our group have demonstrated a high-performance CsPbBr_3_ perovskite PD with low noise, high responsivity, and a fast response speed, showing its significant potential in applications of UWOC [30]. 

In this work, a cost-efficient UWOC system was demonstrated using solution-processed CsPbBr_3_ perovskite PDs as receivers, and based on Verilog hardware description language and FPGA platforms. With a fast response time (rise time of 82 ns, fall time of 710 ns), low noise of 4.3 × 10^−14^ A Hz^−1/2^, and high responsivity of 0.317 A/W, CsPbBr_3_ perovskite PD-based UWOC has achieved a transmission rate of 6.25 Mbps with a low bit error rate (BER) of 0.36%. The performance of perovskite PDs in UWOC was compared with a commercial silicon PD with similar device properties to study the factors of PD affecting communication quality. Due to a larger bandgap with an absorption cutoff wavelength of around 530 nm, perovskite PDs exhibit lower background noise under different ambient light intensities. They show better communication stability before reaching the data rate threshold of 6.25 Mbps, while the silicon PD exhibits a gradually increasing BER, which is likely thanks to the slight influence of noise from ambient light for perovskite PDs. The BER of perovskite PDs increases rapidly after the threshold, due to the long fall time causing difficulty in distinguishing switching signals. Therefore, reducing the fall time of perovskite PDs and the adoption of advanced coding methods will help to further improve the performance of UWOC systems based on perovskite PDs. This work not only demonstrates the huge potential of solution-processed perovskite PDs as receivers for a cost-efficient UWOC system, but also provides tips for achieving a higher data rate UWOC system for practical application.

## 2. Materials and Methods

### 2.1. Photodetector Fabrication

Materials and Reagents

Indium tin oxide (ITO) substrates, PbBr_2_, CsBr, and SnO_2_ (12%) hydrocolloid dispersions, and P3HT (poly[3-hexylthiophene-2,5-diyl]), were purchased from Advance Election Technology Co., Ltd. (Shenyang, China). Dimethyl sulfoxide (DMSO) (>99.7%), Chlorobenzene (CB), and isopropyl alcohol (IPA) (>99.5%) were purchased from Acros (Guangzhou, China). Conductive carbon paste was obtained from Shanghai MaterWin New Materials Co., Ltd. (Shanghai, China). SnO_2_ (5%) precursor solution was prepared by diluting SnO_2_ (12%) hydrocolloid dispersions with deionized water. For the precursor solution of the hole extraction layer, P3HT was dissolved in CB with a concentration of 10 mg/mL. An intermediate precursor solution was prepared by mixing 1.5 mmol PbBr_2_ and 0.6 mmol CsBr in 1 mL DMSO, which was stirred overnight until its was completely dissolved. A high-concentration CsBr solution was prepared by dissolving 0.9 mmol CsBr with IPA and water mixing solvent (6:4 in volume).

Device Fabrication

ITO conductive substrates (2 cm × 2 cm) were cleaned by ultrasonic sequentially in deionized water, acetone, and absolute ethanol for 20 min, respectively. The cleaned substrates were dried with nitrogen, followed by a 20 min UV-ozone treatment before being used. In the first step of fabrication, a thin layer of SnO_2_ was deposited by spin-coating a 5% diluted SnO_2_ precursor solution at 4000 rpm for 30 s, then thermally annealing it at 180 °C for 30 min in ambient air. After cooling down, the ITO/SnO_2_ substrates were transferred to a glovebox with inert gasses. The intermediate solution was spin-coated on the ITO/SnO_2_ substrates at 2000 rpm for 100 s, and 150 μL of CB was used to wash the intermediate film in the last 20 s. Then, the perovskite intermediate films were annealed at 100 °C for 30 min. Next, 100 μL of CsBr solution was dropped onto the intermediate films, which was spun at a speed of 3000 rpm for the last 10 s, and then annealed at 250 °C for 20 min to obtain CsPbBr_3_ perovskite films. Before depositing the top electrode, a thin hole extraction layer of P3HT was spin-coated on the perovskite. Finally, the top carbon electrode was screen-printed on the CsPbBr_3_ films with a conductive carbon paste and annealed at 100 °C for 15 min, and then a 200-nm copper was thermally evaporated atop carbon electrode.

Characterizations

The morphologies of the CsPbBr_3_ films were observed using a scanning electron microscope (SEM, CrossBeam 350 FEI) at an operating voltage of 3 kV. The photoluminescence (PL) spectra were measured using the FluoTime 300 system from Pico Quant (Berlin, Germany) and excited by a picosecond pulsed laser with a wavelength of 375 nm. The current-to-voltage (I–V) characteristics were measured using a Keithley 2635B, and the noise spectrum was obtained by fast Fourier transformation of the dark current. Responsivity (*R*) was measured using a QE-R quantum efficiency system from Enli Technology Co., Ltd. (Shanghai, China), which was calibrated with a standard Si photodetector. The rise and fall times of the photodetectors were measured using a digital oscilloscope with a semiconductor laser (485 nm) in a square wave triggered by a function generator.

### 2.2. Underwater Wireless Optical Communication System Construction

Physical Implementation

The UWOC system consists of a transmission section, a 0.3 m water tank with sea water, and a reception section, as shown in Figure 1. In the transmission section, the FPGA board connected to the host PC via a serial port. Since the communication between the FPGA board and the host PC was of a lower data rate than the transmission speed of the optical part, an IP core-generated FIFO was used for data flow buffering. A TLV3501 comparator was employed as the driving circuit to drive the optical transmitter, which provided sufficient driving capability suitable for different light sources. In our case, the light source was a 488 nm semiconductor laser. In the reception section, the PDs worked in a photovoltaic mode, and an OPA855 transimpedance amplifier was adopted to convert the small photocurrent signal to an amplified voltage signal. Before the demodulation, the obtained voltage analog signals were converted to digital signals using an AN9238 analog to digital converter (ADC) module. The FPGA board in the reception section demodulated the input signal and transferred the received messages to the host PC. For a comparison of the PDs with different bandgaps in optical filter-free UWOC, a commercial silicon PD (S1226-18BK) from HAMAMATSU with a light-sensitive area (1.21 mm²) close to our device was selected, with a maximum dark current of 2 pA, a peak responsivity of 0.36 A/W at 720 nm, and a rise time of 150 ns [31].

Working Mechanism

The FPGA board of transmission section received instructions and messages from the host PC and then modulated the messages using an On-Off Keying (OOK) encoding method. In the following, the FPGA controlled the driving circuit of the optical emitter to modulate the optical signal intensity for data transmission that was based on the RS-232 serial data transmission standard. In this step, a state machine was used to generate transmission signals, as shown in Figure 2a. Upon receiving a transmission signal, the sending module state changed from the “S_IDLE” to the “S_START” state, in which the voltage of the emitter was pulled to a low level for one symbol period. In the next moment, the sending module entered the “S_SEND_BYTE” state, where the data in buffer were read and the signal level was changed sequentially over eight symbol periods. When the sending process was completed, the system would enter the “S_STOP” state, pulling up the level for one symbol period, and then returning to “S_IDLE” if there were no more tasks in the sequence.

In the reception section, when the optical signals reached the PD, they were first converted into small electrical signals. The transimpedance amplifier connected to the PD output amplified the small photocurrent signals to large voltage signals that were further converted into digital signals by an ADC module. The FPGA board demodulated the digital signals into messages. The obtained messages were then handled by the receiving module to send them to the host PC through a state machine, as shown in Figure 2b. After powering up, the receiver module defaulted to a receiver-enabled state that continuously read the input signal, called the “S_IDLE” state. When it read a falling edge of a start bit, indicating the start of transmission, the receiver module would change to the “S_START” state. After one symbol period, it changed to the “S_REC_BYTE” state and began receiving data, where eight bits of data were received over eight symbol periods. To satisfy the sampling theorem, and to receive data accurately, each data bit was sampled at the midpoint of the symbol period. Once it received the stop bit indicating completion of data reception, the receiver module would enter the “S_STOP” state. To avoid missing the start bit of the next byte, it waited only half a symbol period before turning to the “S_DATA” state, in which the received data were transferred to the FPGA, and storing it in the FIFO before sending it to the host PC via the serial port.

In the data transmission tests, Python was utilized to generate 65,535 random data points with values ranging from 0 to 255, and these data were stored in a test data text file. To test system’s stability at the highest on/off switching frequency, data with the feature of “10” in binary were generated using Python. In binary representation, the number 85 is “01010101”. According to the RS-232 frame format, the number “85” was transmitted in the form of “10101010”, with a cyclic pattern of “0101010101” when including the start and stop bits, which represents the scenario with the highest on/off switching frequency. In the image transmission test, Python was used to process the original image into a 256 × 256-pixel grayscale image and a 128 × 128 RGB image, with a total of 65,536 and 49,152 pixels, respectively. The test program read the processed images and sent them to the transmission section. The data were then sent through the optical channel and obtained from the receiving section to the PC. The received data were compared with the transmitted data to calculate the bit error rate, which was used to assess the transmission performance.

## 3. Results and Discussion

### 3.1. The Performance of Solution-Processed Perovskite PDs

CsPbBr_3_ perovskite polycrystalline films were prepared by a two-step solution method proposed by our group previously [30]. The SEM image in Figure 3a is the morphology of the perovskite films, showing their compact and uniform features with a grain size close to 1 μm. The CsPbBr_3_ perovskite film was further characterized with a PL spectrum as shown in Figure 3b. The sole PL peak in the spectrum indicates the pure phase of CsPbBr_3_ perovskite, and the peak location at 525 nm is consistent with its bandgap of about 2.3 eV [32]. Perovskite PDs were fabricated based on high-quality film with a device figuration of ITO/SnO_2_/CsPbBr_3_/P3HT/Carbon, as shown in Figure 4a. The I–V characteristic curves of the PDs were measured with or without the illumination of 480 nm light, as shown in Figure 4a. When the PD is operated in self-power mode (photovoltaic mode), the photocurrent is 6.5 × 10^−6^ A and the dark current is as low as 6.7 × 10^−12^ A, resulting in an on/off ratio of about 10^6^. Pinholes in the perovskite layer can cause serious current leakage, resulting a high dark current. Such a low dark current of our PDs is attributed to the high-quality CsPbBr_3_ films, which are comparable to the commercial silicon PDs with same active area.

There are several key parameters in evaluating the performance of PDs. Noise current is one of the crucial parameters of PDs, which generally consists of shot noise (*i_shot_*) and thermal noise (*i_thermal_*), that can be expressed as follows [33]:(1)$$i_{shot}=\sqrt{2eI_{dark}B},$$
(2)$$i_{thermal}=\sqrt{\frac{4k_{B}TB}{R_{es}}},$$
(3)$$i_{n}=\sqrt{i_{shot}^{2}+i_{thermal}^{2}},$$
where *e* is the elementary charge, *I_dark_* is the dark current, and *B* is the bandwidth, *k_B_* is the Boltzmann constant, *T* is the absolute temperature, and *R_es_* is the resistance of the PDs, which can be calculated from the I–V curve at 0 V as 1.78 × 10^7^ Ω. The shot noise and the thermal noise calculated from Equations (1) and (2) are about 1.5 × 10^−15^ A Hz^−1/2^ and 3.1 × 10^−14^ A Hz^−1/2^, respectively. Since the thermal noise is an order of magnitude higher than the shot noise, indication of the noise current of these perovskite PDs is dominated by thermal noise. The noise current was also estimated by extracting from the fast Fourier transform of the dark current, monitored for 200 s, as shown in Figure 4b. The measured noise current (4.3 × 10^−14^ A Hz^−1/2^) is close to the calculated thermal noise. Responsivity (*R*) and specific detectivity (*D**) are also key figure-of-merit parameters of PDs. *R*, indicating how efficiently the detector converts the optical signal to an electrical signal, is the ratio of photocurrent to incident-light intensity. *D** characterizes the ability to detect how weak the light is, determined by the responsivity and the noise. R and *D** can be calculated as follows [34]:(4)$$R=\frac{I_{ph}}{P_{in}}=\frac{I_{light}-I_{dark}}{P_{in}},$$
(5)$$D^{\text{*}}=\frac{R\sqrt{AB}}{i_{n}}\;,$$
where *I_ph_* is the photocurrent and *P_in_* is the incident-light intensity, *I_light_* and *I_dark_* are the current with and without light illumination, and *A* is the device area. As shown in Figure 4c, CsPbBr_3_ perovskite PDs can respond from 300 to 550 nm, and have a top *R* of 0.317 A/W at 510 nm. The high *R* and low *i_noise_* contribute to a high *D** over 10^12^ Jones in the spectrum range from 380 to 520 nm and a peak value of 1.47 × 10^12^ Jones. The response speed is another crucial parameter of PDs for UWOC, including the rise time and fall time, which are defined as the time differences between the points at which the PD reaches 10% and 90% of its peak response. The rise time and fall time were characterized by the photocurrent response under square-wave modulated laser illumination. As shown in Figure 4d, perovskite PDs exhibit a rise time of only 82 ns and a fall time of 0.71 μs, with a sensitive area of about 1 mm^2^, which is comparable to the silicon PD, with a similar device area. The response time of perovskite PDs is generally governed by the RC time constant [35], since high-quality perovskites device have high carrier mobility and small thicknesses. In our case, when the device area decreases from 4 mm^2^ to 1 mm^2^, the rise time and fall time decrease from 345 ns and 1.37 μs to 82 ns and 0.71 μs, respectively, indicating that the response time of our PDs is mainly governed by the RC constant. The high response speed and responsivity of CsPbBr_3_ PDs suggest their significant potential in the application of UWOC.

### 3.2. The UWOC Application

To test the performance of data transmission, the waveform of obtained data from the 12-bit ADC was recorded and marked as “received_data” as shown in Figure 5a. A binary digital waveform was obtained via voltage comparison and named “decided_data”. Due to the noise of the PDs and interferences from the environment, decision errors may occur at certain points where the voltage signal fluctuation happens in the level close to the decision threshold, leading to glitches in the “decided_data” waveform as highlighted in Figure 5a by red frame. This kind of glitch can seriously reduce the robustness of a UWOC system. They can cause data decision errors when they appear within data bits, and cause the premature reception or misalignment of subsequent data when they occurs outside of data bits. To deal with this issue, a filtering operation was introduced by caching the waveform data for one clock cycle. The “0/1” switching is effective only when the later received data match with the cached data, otherwise they would be treated as jitter. This operation can filter out glitches lasting for one clock cycle, reducing the decision error rate and improving communication quality. As shown in Figure 5a, the glitch in the “decided_data” was removed, and correct data was obtained in “final_data”.

The bit error rate (BER) is a key performance parameter of a communication system. Figure 5b shows the BER of transmitting random data at different data transmission rates for both perovskite PDs and silicon PDs. The BER of perovskite PDs exhibits a steeper curve compared to the silicon PDs, indicating that the perovskite PDs show more stable communication performance before reaching their frequency limit (6.25 Mbps), although silicon PDs display better performance in their high data-transmission rate. The BER of transmitting repeated “01” for the frequent on/off switching scenario is shown in Figure 5c. It indicates that the perovskite PD demonstrates better stability compared to the silicon PD in frequent “01” transitions. An analysis of data errors shows that the errors of silicon PD are mainly from the misinterpretation of “0” as “1”, for example, the number 85 (“01010101”) being misinterpreted as 87 (“01010111”) or 255 (“11111111”). In contrast, the errors in the perovskite PD tend to be missing data packets or missing some data points, which is easily observed in subsequent image transmissions. To better understand the difference in BER between perovskite PDs and silicon PDs, the waveforms of perovskite PDs and silicon PDs under the illumination of high-frequency optical signals were investigated by a digital oscilloscope. As shown in Figure 6a,b, the silicon PD exhibits a narrower response than the perovskite PD. The ratios of time spans for high voltage levels and low voltage levels were calculated, which theoretically are one due to the on/off of optical signals occupying equal time spans. The ratio of the silicon PD is only 0.29, far less than that of the perovskite PD (0.71), indicating that the perovskite PD has fewer chances to cause misinterpretation. The background currents and noise currents were also studied under different environment illuminations, which were simulated using a Xe Lamp and a neutral-density filter. As shown in Figure 6c,d, perovskite PDs exhibit lower background currents and noise currents in the environment light condition, even though the silicon PD has lower intrinsic noise and dark current. The lower noise of the perovskite PD should be the reason that contributes to the stable communication performance.

To demonstrate image transmission via the UWOC system, a grayscale image of 256 × 256 pixels and a RGB image of 128 × 128 pixels were first generated using Python, with a total of 65,536 and 49,152 pixels, respectively. Figure 7 shows the changes in the images after transmission at a data rate of 7.14 Mbps and 8.33 Mbps for the grayscale and RGB images, respectively. From these images, one can easily perceive that the errors in the images received by the silicon PD are mainly the difference in gray value and color, while that of the perovskite PD are pixel offset and deformation. This difference in image distortion is attributed to different error mechanisms for silicon and perovskite PDs. The bit error rate is typically not suitable for assessing image quality due to its disregard for visual perception. To better assess the fidelity of the images after transmission, mean squared error (MSE) and structural similarity index (SSIM) are introduced. The MSE takes into account the difference in each pixel to reflect the pixel divergence of an image, and the SSIM considers contrast, brightness, and structure of the image, providing a more accurate assessment of image fidelity, especially in contexts involving human perception. The calculated MSE and SSIM are plotted in Figure 8. Similar to the text transmission, the perovskite PD exhibits better fidelity than the silicon PD before reaching a limit. Once the perovskite PD operated above the data rate threshold, the image fidelity decreased rapidly and was worse than that of the silicon PD, which is due to the data losses of the perovskite PD leading to entire data misalignment. For the perovskite PD, the errors in the pixel are relatively few, and the grayscale and color information in the images are of good fidelity. This kind of error from packet loss can potentially be avoided by using automatic repeat request (ARQ) and forward error correction (FEC) coding methods [36,37]. Conversely, the images from the silicon PD mainly have the most errors in the pixel itself, with brighter and noticeable white spots and no obvious region misalignment. This can be ascribed to the misinterpretation of the “0” (corresponding to a high voltage level in the waveform, where the light source is on) due to the time span of the high voltage gradually reducing with increasing data rate. When the “0” of a pixel is misinterpreted as “1” in every case, the data become “255”, resulting in a white spot in the image.

## 4. Conclusions

In this work, high-performance all-inorganic perovskites PDs were fabricated using a solution process approach. Based on these perovskite PDs, a cost-efficient underwater wireless optical communications system was successfully demonstrated with an FPGA platform, achieving a data rate of 6.25 Mbps with low bit error rate of 0.36%. Since the large bandgap with an absorption cutoff at around 530 nm leads to no absorption of long-wavelength light, perovskite PDs have lower background noise under different intensities of environment illumination. They exhibit better communication stability before reaching a data rate threshold of 6.25 Mbps, while the silicon PDs show a growing BER with increasing communication rate. However, the long fall time of perovskite PDs results in difficulty in distinguishing switching signals at high data rates, causing the BER to increase rapidly. Reducing the fall time of perovskite PDs and using advanced coding techniques can help to further improve the performance of the UWOC system based on perovskite PDs. This work not only demonstrates the potential of perovskite PDs in the application of UWOC, but also benefits the development of a cost-effective UWOC system based on FPGAs.

## Figures and Tables

**Figure 1 micromachines-15-01185-f001:**
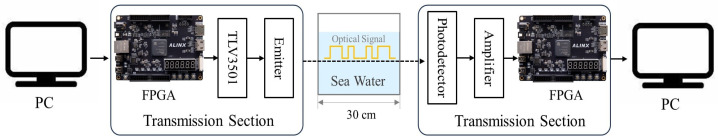
The block diagram of the UWOC system.

**Figure 2 micromachines-15-01185-f002:**
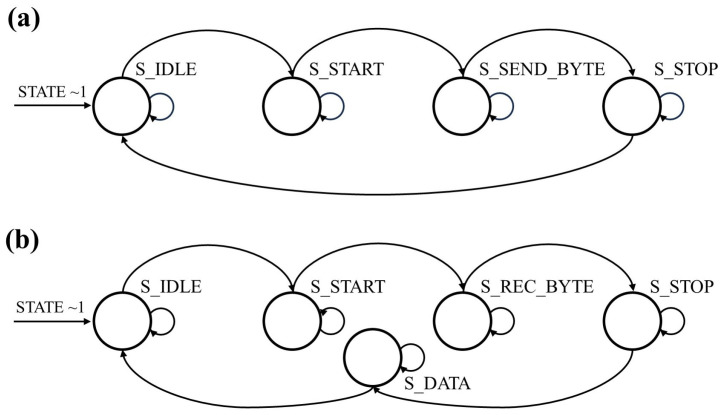
The working mechanism of state machine for (**a**) transition; (**b**) reception.

**Figure 3 micromachines-15-01185-f003:**
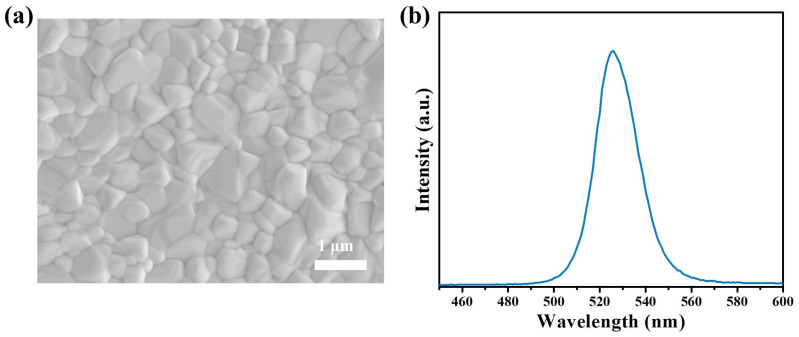
(**a**) SEM image; (**b**) steady-state photoluminescence spectra of the solution processed CsPbBr_3_ perovskite films.

**Figure 4 micromachines-15-01185-f004:**
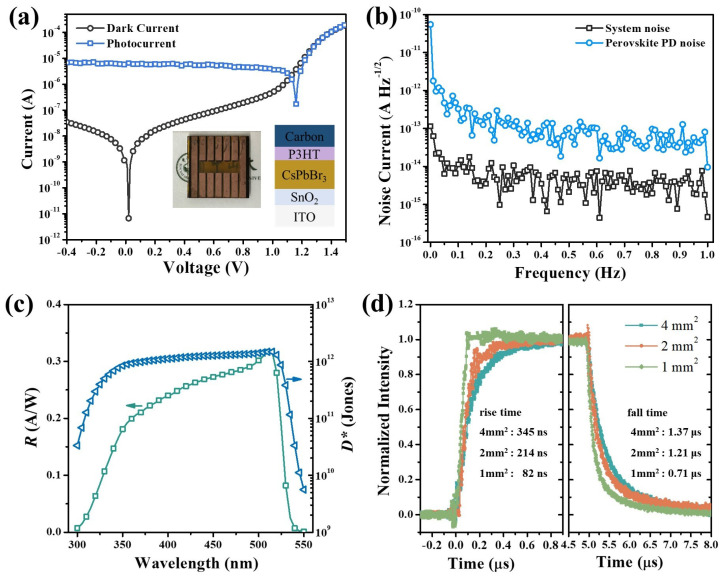
(**a**) I–V characteristics of the perovskite PDs with and without illumination. The insert is the schematic and optical image of the fabricated CsPbBr_3_ PD. (**b**) The measured noise current spectra of the instrument and CsPbBr_3_ perovskite PDs at the photovoltaic mode, (**c**) Responsivity (in green) and specific detectivity (in blue) spectrum of perovskite PDs. (**d**) Transient response of perovskite PDs with different sensitive areas under the illumination of a 488 nm laser at a frequency of 100 kHz.

**Figure 5 micromachines-15-01185-f005:**
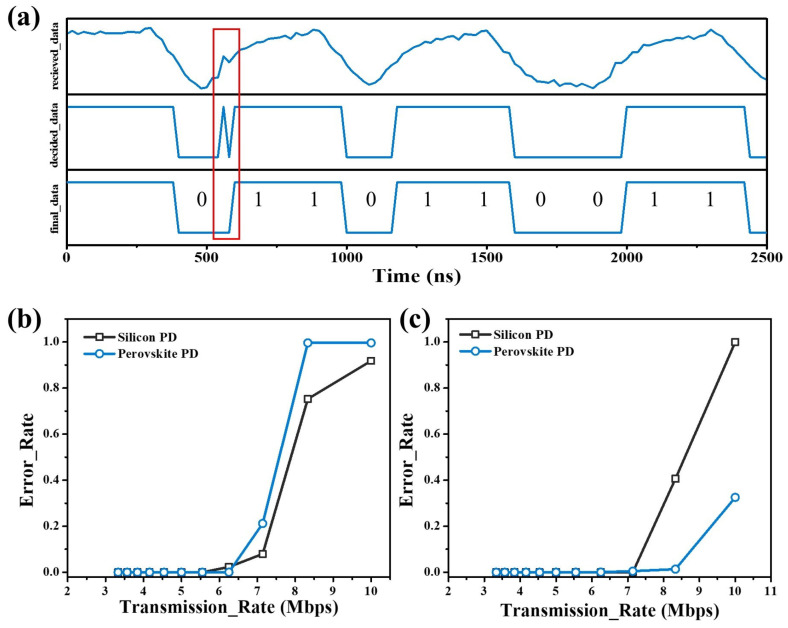
(**a**) The waveform of received signals in the raw, decided, and final states. The red frame highlights the glitches; the BER of PDs for different transmission rates of (**b**) random data and (**c**) regular “01” data.

**Figure 6 micromachines-15-01185-f006:**
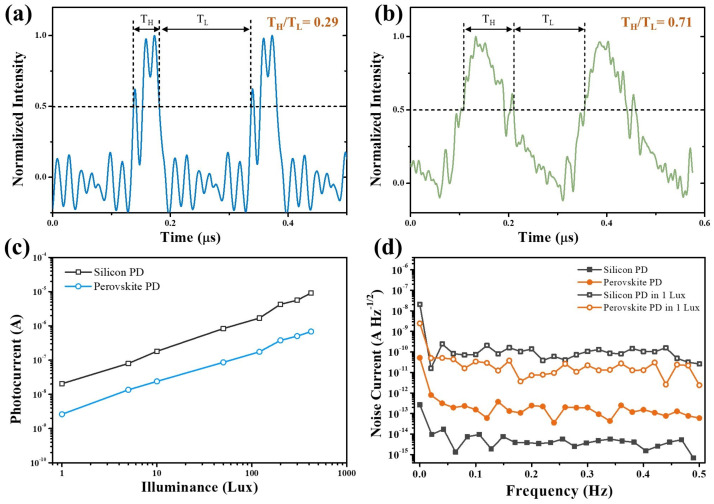
The waveform of the transient response of (**a**) the silicon PD and (**b**) the perovskite PDs; (**c**) the background photocurrent of the PDs under different intensities of environment illumination; (**d**) noise current of silicon and perovskite PDs under dark and 1 lux white light illumination.

**Figure 7 micromachines-15-01185-f007:**
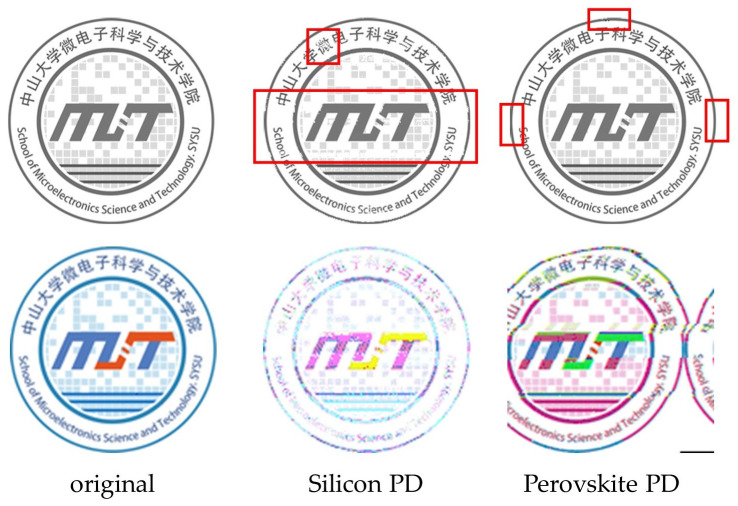
The change in images after transmission at 7.14 Mbps for grayscale the images and 8.33 Mbps for the RGB images.

**Figure 8 micromachines-15-01185-f008:**
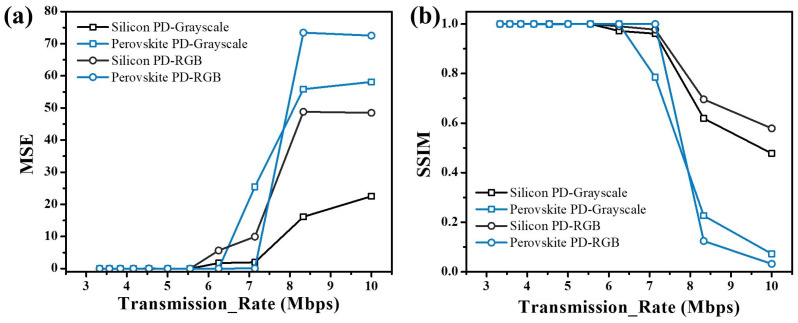
The changes in (**a**) MSE and (**b**) SSIM for the silicon and perovskite PDs at different transmission rates.

## Data Availability

The data provided in this study are available upon request from the corresponding author.

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
