# Peer review of "Solution-Processed CsPbBr3 Perovskite Photodetectors for Cost-Efficient Underwater Wireless Optical Communication System"

_micromachines, 2024, doi:10.3390/mi15101185_

Round 1

Reviewer 1 Report

Comments and Suggestions for Authors

Referee report: “Solution Processed CsPbBr3 Perovskite Photodetectors for Cost- Efficient Underwater Wireless Optical Communication System

This is a quite interesting research paper that probably can be recommended for publication, but only after clarifying and detailing some parts of the text.

1.     Introduction. First paragraph. Lines 29-43.  More supporting references are needed, since it is not clear where the authors' original inventions are, and where those are borrowed from literature and require supporting references. Reference [1] although not so ancient, but what has been done in the world over the last 5 years was interesting to highlight.

2.     Lines 45 and 50. The end of this sentences needs supporting references.

3.     Line 65. It would be useful to give a value for band gap energy

4.     Line 210. The value Eg=2.3 eV given here contradicts the earlier statement that CsPbBr3 is a wide-band material.

5.     Fig.3. No information is given on the photoluminescence excitation wavelength.

6.     Did you check the aging of fabricated CsPbBr3 perovskites?

7.     Has aging been checked for the resulting CsPbBr3 perovskite structures?

In general, the manuscript is interesting and can be recommended for publication after constructive reflection on the above comments.

Reviewer 2 Report

Comments and Suggestions for Authors

Dear Authors,

Please put a schematic of your device and an optical image

You write "Next, 100 μL of CsBr solution was dropped onto the intermediate films, which was spun
at a speed of 3000 rpm for the last 10 s, and then annealing at 250 °C for 20 min to obtain
CsPbBr3 perovskite films" This is wrong, it is not CsBr. Which of the high or low concentration did you use. same results?
